# The Efficacy of Fibrinogen Concentrates in Relation to Cryoprecipitate in Restoring Clot Integrity and Stability against Lysis

**DOI:** 10.3390/ijms23062944

**Published:** 2022-03-09

**Authors:** Claire S. Whyte, Akriti Rastogi, Ellis Ferguson, Michela Donnarumma, Nicola J. Mutch

**Affiliations:** Aberdeen Cardiovascular and Diabetes Centre, Institute of Medical Sciences, School of Medicine, Medical Sciences and Nutrition, University of Aberdeen, Foresterhill, Aberdeen AB25 2ZD, UK; c.s.whyte@abdn.ac.uk (C.S.W.); a.rastogi@oxb.com (A.R.); ellis.ferguson@nhs.scot (E.F.); m.donnarumma@abdn.ac.uk (M.D.)

**Keywords:** fibrinogen, fibrinolysis, coagulation, trauma, cryoprecipitate

## Abstract

Loss of fibrinogen is a feature of trauma-induced coagulopathy (TIC), and restoring this clotting factor is protective against hemorrhages. We compared the efficacy of cryoprecipitate, and of the fibrinogen concentrates RiaSTAP^®^ and FibCLOT^®^ in restoring the clot integrity in models of TIC. Cryoprecipitate and FibCLOT^®^ produced clots with higher maximal absorbance and enhanced resistance to lysis relative to RiaSTAP^®^. The fibrin structure of clots, comprising cryoprecipitate and FibCLOT^®^, mirrored those of normal plasma, whereas those with RiaSTAP^®^ showed stunted fibers and reduced porosity. The hemodilution of whole blood reduced the maximum clot firmness (MCF) as assessed by thromboelastography. MCF could be restored with the inclusion of 1 mg/mL of fibrinogen, but only FibCLOT^®^ was effective at stabilizing against lysis. The overall clot strength, measured using the Quantra^®^ hemostasis analyzer, was restored with both fibrinogen concentrates but not cryoprecipitate. α_2_antiplasmin and plasminogen activator inhibitor-1 (PAI-1) were constituents of cryoprecipitate but were negligible in RiaSTAP^®^ and FibCLOT^®^. Interestingly, cryoprecipitate and FibCLOT^®^ contained significantly higher factor XIII (FXIII) levels, approximately three-fold higher than RiaSTAP^®^. Our data show that 1 mg/mL fibrinogen, a clinically achievable concentration, can restore adequate clot integrity. However, FibCLOT^®^, which contained more FXIII, was superior in normalizing the clot structure and in stabilizing hemodiluted clots against mechanical and fibrinolytic degradation.

## 1. Introduction

Uncontrolled hemorrhage is a preventable consequence of severe trauma, which accounts for more than 5 million deaths worldwide annually [1]. The resulting trauma-induced coagulopathy (TIC) is characterized by severe dysfunction of the hemostatic system, giving rise to acute bleeding during the early phase post-injury. There have been numerous names given to TIC over the years, reflecting the differing clinical manifestations and complex pathophysiology of this disorder, which is not fully understood [1,2]. Loss of the coagulation factors contributes to this process, with fibrinogen being more sensitive to depletion than other coagulation factors [3,4]. Hypofibrinogenemia in TIC arises due to hemodilution (from blood product administration), blood loss, consumption during clot formation, hypothermia (which impairs fibrinogen synthesis), fibrinogenolysis and increased degradation due to acidosis [3,4,5,6,7]. A drop in fibrinogen below 1.5 g/L post-surgery gives rise to a significant increase in the risk of post-operative hemorrhage [8,9]. Careful management is therefore required to maintain hemostatic stability post-operatively [10]. Evidence from a prospective study demonstrates that low fibrinogen levels are associated with increased injury severity scores and were an independent predictor of mortality at 24 h and 28 days [11].

Current guidelines recommend fibrinogen replacement during major bleeding when fibrinogen levels drop below 1.5 mg/mL [12,13]. Cryoprecipitate, a pooled blood product derived from fresh frozen plasma, is commonly used within the UK and other countries to increase fibrinogen levels in acute settings. Cryoprecipitate contains additional clotting factors, including von Willebrand factor, factor VIII and factor XIII (FXIII), which may augment clot formation and stabilization. CRYOSTAT was a UK feasibility trial that demonstrated that supplementation with cryoprecipitate in TIC could be achieved within 90 min of admission, with a median time of 60 min [14]. Despite not being powered for secondary end-points, the trial indicated that raising fibrinogen levels in TIC may reduce mortality [14]. CRYOSTAT-2 is a multi-center randomized controlled trial that has recently finished recruiting patients and aims to determine whether cryoprecipitate incorporated into a major hemorrhage protocol can improve survival [15].

Significant disadvantages of cryoprecipitate include batch-to-batch variation, a lack of standardization, an inability to estimate exact fibrinogen concentration, the administration of large volumes and the preparation time in an acute setting. To recover fibrinogen levels to 0.6–1.2 g/L in a 70 kg adult, 10–20 units (0.5–1 L) of cryoprecipitate were required [16]. In contrast, fibrinogen concentrates are dispensed as a standard dose of highly concentrated fibrinogen, which are significantly quicker to reconstitute and administer [17]. This may be critical during acute hemorrhage in terms of timing, availability, and administration. However, there have been conflicting results in the feasibility of administering fibrinogen concentrates. The Canadian Fibrinogen in the Initial Resuscitation of Severe Trauma (FiiRST) trial demonstrated that the majority of patients were able to receive the concentrate within 60 min of admission and that their plasma fibrinogen concentration was increased [18]. However, the UK-based Early Fibrinogen concentrate therapy for major hemorrhage In Trauma (E-FIT 1) trial found that it was not feasible to administer within 45 min, although fibrinogen levels were significantly elevated in the treatment group [19]. The Fibrinogen Early in Severe Trauma StudY (FEISTY) recently directly compared the time to administration of RiaSTAP^®^ (CSL Behring) versus cryoprecipitate [17]. Fibrinogen replacement during hemorrhaging was guided by ROTEM analysis of the clot amplitude at 5 min (FIBTEM A5). This study demonstrated that the fibrinogen concentrate could be administered within 30 min whilst cryoprecipitate required 60 min [17].

Fibrinogen supplementation with concentrates in congenital hypo- or afibrinogenemia have demonstrated a favorable safety profile [20] and can be used as prophylactic regimen in the prevention against recurrent bleeding episodes [21]. Two fibrinogen concentrates currently licensed in the UK for management and perioperative prophylaxis of these conditions are RiaSTAP^®^ (CSL Behring) and FibCLOT^®^ (LFB). Neither of these fibrinogen concentrates are currently licensed for acquired bleeding in the UK but have a favorable safety profile, making them an attractive alternative to cryoprecipitate to restore fibrinogen loss in TIC. Here, we utilized the models of TIC to compare the efficacy of cryoprecipitate to fibrinogen concentrates in restoring clot integrity and stability against fibrinolytic degradation.

## 2. Results

### 2.1. Fibrinogen Supplementation Augments Clot Formation and Stability against Lysis

Fibrinogen-deficient plasma (FDP) was used to mimic loss of fibrinogen in TIC and to compare the ability of cryoprecipitate and fibrinogen concentrates in stabilizing plasma clots. As expected, in the absence of added fibrinogen, no clots formed. The concentrations of fibrinogen above 1 mg/mL were required to observe adequate clot formation, with no discernible increase in absorbance observed with 0.5 mg/mL RiaSTAP^®^ (Figure 1A–C). An extremely strong correlation was observed between the fibrinogen concentration and the maximal absorbance of clots with all sources (Figure 1D). However, the clots formed following the addition of RiaSTAP^®^ exhibited significantly lower maximum absorbances than clots formed after the addition of cryoprecipitate or FibCLOT^®^ (Figure 1D). Even at the highest concentration of fibrinogen tested (2 mg/mL), the maximum absorbance values achieved with all three sources were significantly lower than those observed in pooled normal plasma (PNP) (0.41 ± 0.004). An analysis of the slope of clot formation, indicative of the rate of fibrin polymerization, demonstrated a superior effect of FibCLOT^®^ that was comparable with PNP even at 1.5 mg/mL (Figure 1E).

The addition of FibCLOT^®^, RiaSTAP^®^ or cryoprecipitate added at 1 mg/mL or higher significantly enhanced clot stability and generated 50% lysis times that were equivalent to or higher than that observed with PNP (Figure 2A–F). The strong concentration dependence was much more apparent with both cryoprecipitate and FibCLOT^®^ (Figure 2G), indicative of increased resistance against fibrinolysis degradation.

### 2.2. The Fibrinogen Source Impacts on Fibrin Network Structure

The structure of the fibrin network greatly influences susceptibility to lysis. Confocal microscopy revealed that the supplementation of FDP with fibrinogen increased the fiber and network density. Supplementation with RiaSTAP^®^ produced a fibrin network comprised of stunted fibrin fibers with evidence of increased branching and reduced porosity in a concentration-dependent manner (Figure 3A,B). In marked contrast, homogenous networks of elongated fibers were evident in clots supplemented with cryoprecipitate or FibCLOT^®^ that closely mimicked the structure of the control clot formed from PNP (Figure 3A,C,D). FibCLOT^®^ at 2 mg/mL produced a fibrin network structure most comparable with PNP in terms of fiber length, branching and porosity (Figure 3A,D). These data highlight that different fibrinogen preparations give rise to clots of very different quality and structure that may influence their stability against mechanical stress and fibrinolytic degradation.

### 2.3. Additional Factors Contained within Fibrinogen Sources

FXIII was quantified in the different fibrinogen sources, PNP and FDP. PNP contained 31.5 ± 3.2 μg/mL FXIII, while within FDP, it was reduced to 12.9 ± 0.4 μg/mL. RiaSTAP^®^ contained significantly less FXIII/mg of fibrinogen of all the fibrinogen sources, with 2.5-fold and 2.9-fold less compared with cryoprecipitate and FibCLOT^®^, respectively (Figure 4A). In FDP, α_2-_antiplasmin (α_2_AP) levels were reduced around three-fold compared with PNP (23.9 ± 5.2 vs. 73.1 ± 10.7 μg/mL). RiaSTAP^®^ and FibCLOT^®^ contained negligible levels of α_2_AP (0.4 ± 0.1 and 0.5 ± 0.1 μg/mg of fibrinogen, respectively, Figure 4B), while the cryoprecipitate levels were similar to PNP. Similarly, cryoprecipitate contained high levels of plasminogen activator inhibitor-1 (PAI-1) (2.02 ± 0.01 ng/mL), while fibrinogen concentrates contained very little PAI-1 (0.09 ± 0.01 RiaSTAP^®^; 0.02 ± 0.00 ng/mL FibCLOT^®^).

### 2.4. Fibrinogen Supplementation Recovers Clot Strength in a Hemodilution Model

Our data thus far demonstrate the ability of fibrinogen sources to enhance clot stability in a plasma-based model. To investigate the influence of the cellular components of blood, we used a hemodilution model, in which blood was diluted by 40% to recapitulate blood loss in a trauma setting. Thromboelastography revealed that 40% hemodiluted clots exhibit reduced clot strength (MCF; Figure 5A–C). Supplementation with 1 mg/mL of FibCLOT^®^ or RiaSTAP^®^ was sufficient to normalize the MCF value in hemodiluted clots, whereas 2 mg/mL of cryoprecipitate was required (Figure 5A–C). The degree of clot firmness in the 40% HD clots was directly proportional to the concentration of the added fibrinogen (Figure 5D).

ROTEM analysis was then performed with the inclusion of tPA. Representative traces show that 40% hemodiluted clots lyse more readily than whole blood clots and could be stabilized against fibrinolytic degradation by fibrinogen supplementation (Figure 6A). The addition of fibrinogen normalized the clot formation time, with fibrinogen concentrates being more effective than cryoprecipitate (Figure 6B). A concentration of 2 mg/mL fibrinogen was required to normalize the MCF value in the presence of tPA (Figure 6C). All fibrinogen sources were equally effective at recovering clot elasticity (Figure 6D), with 2 mg/mL being more effective than 1 mg/mL. Interestingly, FibCLOT^®^ was far more effective at increasing the resistance of clots to tPA-mediated fibrinolytic degradation than cryoprecipitate or RiaSTAP^®^ (Figure 6E). The ability of FibCLOT^®^ to stabilize against lysis may arise due to an increase in the crosslinking of α_2_AP into the forming clot by the high levels of FXIII present in this fibrinogen preparation (Figure 4).

Clot strength and stability against lysis were also investigated using a Quantra Hemostasis Analyzer, which uses SEER Sonorheometry to measure the shear modulus of whole blood during coagulation. This methodology provides information on both the platelet and fibrinogen contribution to clot strength. Hemodilution of clots reduced overall clot strength to 58.3% of the whole blood from 18.9 ± 0.4 to 11.0 ± 0.5 hPa (Figure 7A, *p* < 0.001), while the fibrinogen contribution to clot strength reduced to 0.7 ± 0.1 from 1.5 ± 0.2 (Figure 7C). The addition of RiaSTAP^®^ and FibCLOT^®^ recovered the overall clot strength (17.5 ± 1.1 and 18.1 ± 1.1 hPa, respectively, *p* < 0.001) but cryoprecipitate was not as effective. However, all fibrinogen sources restored the fibrinogen contribution to clot strength, with both fibrinogen concentrates exhibiting increased fibrinogen contribution over the control (Figure 7B). Both RiaSTAP^®^ and FibCLOT^®^ were capable of restoring stability against lysis (Figure 7C).

### 2.5. Fibrinogen Supplementation Enhances Thrombus Stability against Fibrinolytic Degradation

We next assessed the stability of thrombi formed from hemodiluted blood in a model that encompasses physiological shear stress to mimic the vasculature. Fluorescently labelled fibrinogen was incorporated prior to thrombus formation and tPA-mediated lysis monitored as the release of fluorescently labelled fibrinogen degradation products. Thrombi formed from hemodiluted whole blood were more sensitive to fibrinolytic degradation than those formed from whole blood (Figure 8A,B). Fibrinogen supplementation at 1 mg/mL by any source was sufficient to stabilize thrombi against fibrinolytic degradation, most likely by enhancing the fibrin clot structure.

## 3. Discussion

Fibrinogen supplementation in the trauma setting has gained considerable interest in recent years [11,17,19]. In this study, we utilized two different models of TIC to compare the impact of different fibrinogen sources on clot integrity and resistance to fibrinolysis. Our data demonstrate that the inclusion of 1 mg/mL fibrinogen, a clinically achievable concentration, restores adequate clot formation. We found that fibrinogen preparations significantly increased turbidity in the FDP model, with FibCLOT^®^ and cryoprecipitate producing more robust curves and maximum absorbances. Resistance to tPA-mediated fibrinolysis was directly proportional to the fibrinogen concentration. Confocal microscopy revealed that clots formed with FibCLOT^®^ resembled those formed from PNP or after supplementation with cryoprecipitate. In marked contrast, clots formed with RiaSTAP^®^ showed abnormal structure with stunted protofibril growth. Using the hemodilution model, we showed that both fibrinogen concentrates and cryoprecipitate restore normal hemostatic parameters in terms of clot formation, mechanical strength and elasticity and that similar contributions to overall clot strength were observed using SEER sonorheomtry. FibCLOT^®^ was more effective than other fibrinogen preparations in conferring resistance to tPA-mediated fibrinolysis in thromboelastography and SEER Sonorheometry. Cryoprecipitate is a heterogenous preparation and, in line with this, was shown to contain high levels of PAI-1 and α_2_AP, which were absent in the fibrinogen preparations. Interestingly, the levels of FXIII were exceptionally high in FibCLOT^®^, significantly higher than even that of cryoprecipitate, which may account for the increased mechanical strength and clot stability. This study reveals for the first time in a whole blood model of TIC that fibrinogen supplementation can restore normal clot integrity and strength, thereby promoting resistance to mechanical disruption and fibrinolytic degradation.

Supplementation of 1 mg/mL of fibrinogen in the FDP model was sufficient to promote appreciable clot formation, although the inclusion of cryoprecipitate and FibCLOT^®^ produced higher maximum absorbances. It should be noted that the initial maximum absorbance with cryoprecipitate was dependent on the concentration, reflecting the influence of the color and heterogenous mix of proteins in this supplement. The clot formation slope with FibCLOT^®^ was steeper and more dependent on the fibrinogen concentration, indicative of a faster rate of lateral growth of fibrin fibers and branch points [22]. The changes observed in the clot formation curves and maximum absorbance were reflected in ultrastructure of the fibrin network, where both cryoprecipitate and FibCLOT^®^ produced a fibrin network that is more akin to that observed in normal plasma, with similar fiber length, branching and porosity. In contrast, clots supplemented with RiaSTAP^®^ demonstrated stunted fibers with abnormal clustering or knotted areas, as previously described [23]. It is well established that fibrin structure impacts the rate at which fibrinolysis proceeds [24]. Intriguingly, despite the fact that thin fibers are a better surface for plasminogen activation and lyse faster at an individual level, clots comprising thinner fibers are more resistant to lysis [25,26]. We observed a strong correlation between fibrinogen concentration and resistance to lysis with cryoprecipitate and FibCLOT^®^ most likely reflecting the change in the quality of the network structure with fibrinogen supplementation by these sources.

The different behaviors of the fibrinogen concentrates may reflect differences in the purification procedure. FibCLOT^®^ is recovered from cryosupernatant and precipitated by ion exchange chromatography, whilst RiaSTAP^®^ is manufactured from glycine precipitation of cryoprecipitate, followed by several precipitation/adsorption steps. Aggregates present in fibrinogen preparations have been shown to dramatically alter the fibrin network structure and may therefore account for the differences observed, with FibCLOT^®^ being comprised of fibrin monomers that give rise to long needle-like fibers [27]. Alternatively, excipients used to stabilize the fibrinogen preparations may impact clot formation. For example, RiaSTAP^®^ contains albumin [28], which is known to decrease maximum turbidity and produce thinner fibrin fibers [29]. Increasing ionic strength shortens clotting times, reduces max turbidity, and alters fiber and pore sizes, with the most dramatic changes observed above the normal physiological range [30,31]. FibCLOT^®^ contains a maximum of 69 mg sodium per vial, whilst RiaSTAP^®^ contains 2.5 times more at 164 mg per vial included in the form of sodium chloride. Chloride ions reduce lateral aggregation, producing thinner and more curved fibers [32], which may partially explain the differences in clot structure between these fibrinogen concentrates.

In thromboelastography, the MCF value relates to the absolute strength of fibrin, with a low MCF indicative of decreased fibrinogen and/or changes in fibrin polymerization. We found that fibrinogen concentrates produced more robust MCF values compared to cryoprecipitate, consistent with reported increases in MCF observed post-administration of fibrinogen concentrates for congenital afibrinogenmia [33,34]. Cryoprecipitate was also inferior to both fibrinogen concentrates in restoring clot strength when analyzed using SEER Sonorheometry. The dramatic increase in MCF and clot strength noted with fibrinogen concentrates may arise due to enhanced fibrin polymerization and lateral aggregation, thereby producing a stronger mechanical fibrin network.

Fibrinogen serves as a carrier for the heteromeric FXIII complex in blood [35], and as a consequence, these proteins co-purify. FibCLOT^®^ contained significantly higher levels of FXIII/mg of fibrinogen than cryoprecipitate and almost three-fold higher concentrations than RiaSTAP^®^. These data align with a previous report showing higher FXIII activity in FibCLOT^®^ and a third concentrate, Fibryga^®^, compared with RiaSTAP^®^ [28]. The presence of FXIII during clot formation increases the clot stiffness [36,37,38] and density and reduces the porosity, which augments stability against fibrinolysis [39,40]. FXIII confers further resistance to lysis by crosslinking fibrinolytic inhibitors, predominantly α_2_AP [41,42], into the thrombi. The application of shear stress is necessary to observe the impact of FXIII crosslinking of α_2_AP on fibrinolysis [43]. We found a significant reduction in thrombus stability in our hemodilution model, which was restored to the same degree as all fibrinogen sources. However, we previously demonstrated using a Chandler model thrombus system that 50% of the circulating plasma FXIII level and 60% of α_2_AP level are sufficient to stabilize thrombi against lysis [42,43]. Thus, in this model, we have adequate levels of these factors to support thrombus stability upon fibrinogen supplementation. Of note, we observed differences in the susceptibility to tPA-mediated lysis using both thromboelastography and SEER Sonorheometry, suggesting different sensitivities of these assays to factor levels in plasma.

In this study, we demonstrated that supplementation of 1 mg/mL fibrinogen, a clinically achievable concentration, can restore adequate clot integrity and stability in a whole blood model of TIC. We found differences in the efficacy of the two fibrinogen concentrates analyzed, presumably arising due to differences in their preparation and composition. Indeed, the high levels of FXIII present in FibCLOT^®^ appear to confer increased clot strength and resistance to lysis, over and above RiaSTAP^®^ and even cryoprecipitate. These data indicate that, while FibCLOT^®^ primarily restores fibrinogen, the co-infusion of FXIII, as a by-product of this supplement, augments clot integrity and stability against mechanical and fibrinolytic degradation. These mechanistic studies reveal that several different attributes of various fibrinogen sources may impact the effectiveness of these products in the acute clinical setting.

## 4. Methods

### 4.1. Subjects or Blood Collection

All blood samples were obtained after approval by The University of Aberdeen College Ethical Review Board in accordance with the Declaration of Helsinki and after obtaining written consent. Citrated whole blood was collected from consented healthy subjects in a 0.1 volume of 0.13 M trisodium citrate. Pooled normal plasma was prepared by collecting blood donations from 20 normal healthy volunteers [44]. Plasma was collected following centrifugation at 3000× *g* for 30 min at 4 °C and subsequently pooled, aliquoted and stored at −70 °C until required.

### 4.2. Fibrinogen Depletion Models

Fibrinogen-deficient plasma (FDP) or whole blood diluted with saline (0.9% NaCl) (60:40 blood:saline) was used to mimic trauma conditions [45]. The normal plasma concentration of fibrinogen ranges from 1.5 to 4 mg/mL. Fibrinogen was supplemented into models to give equivalent plasma concentrations of 0.5–3 mg/mL fibrinogen, with the amount dependent on the assay. We compared cryoprecipitate and the fibrinogen concentrates in their ability to stabilize clots by various methods. The fibrinogen concentration in cryoprecipitate was 9.53 mg/mL (kindly measured by Aberdeen Royal Infirmary Haematology department). To simplify and for ease of comparison between techniques, we refer to the equivalent plasma concentration.

### 4.3. Turbidity Assays

Clots were formed from 30% PNP or FDP supplemented with cryoprecipitate, RiaSTAP^®^ (CSL Behring GmbH, Marburg, Germany) or FibCLOT^®^ (LFB, Les Ulis, France) (0.5–2 mg/mL) in the presence of 16 mM phospholipids (Rossix, Mölndal, Sweden) ± 300 pM tPA (Alteplase (Actilyse^®^) Boehringer Ingelheim International GmbH, Ingelheim am Rhein, Germany) in Tris buffered saline with Tween-20 (TBST; 10 mM Tris, 140 mM NaCl and 0.01% Tween-20 pH 7.4). Clotting was initiated with CaCl_2_ (10.6 mM) and thrombin (0.1 U/mL, Sigma Aldrich Sigma-Aldrich, Darmstadt, Germany), and absorbance readings were taken every min for 4 h at 37 °C in a in a FLX-800 plate reader (Biotek Instruments, Winooski, VT, USA).

### 4.4. Thromboelastography

FDP (63%), whole blood or 40% hemodiluted blood was spiked with cryoprecipitate, RiaSTAP^®^ and FibCLOT^®^ (0.5–3 mg/mL), with or without 0.4 nM tPA. The EXTEM assay (using Star-tem^®^ Reagent (recalcifies sample) and Ex-tem^®^ Reagent (concentration of tissue factor and phospholipid) were performed in accordance with manufacturer’s guidelines (ROTEM Tem International GmbH, Munchen, Germany).

### 4.5. Confocal Microscopy

Clots were formed in Ibidi Vi^0.4^ μ slides (Ibidi GmbH, Gräfelfing, Germany), consisting of 30% PNP or FDP supplemented with cryoprecipitate, RiaSTAP^®^ or FibCLOT^®^ (0.5–2 mg/mL) in the presence of 16 mM phospholipids (Rossix) and AF488-fibrinogen (0.25 µM) in TBST. Clotting was initiated with CaCl_2_ (10.6 mM) and thrombin (0.1 U/mL), and the clots were allowed to form for 2 h at 37 °C. The images were recorded on Zeiss 710 laser scanning confocal microscope with a 63 × 1.40 oil immersion objective using Zeiss Zen 2012 software, (Carl Zeiss Ltd, Jena, Germany).

### 4.6. Determination of Plasma Protein Concentrations

The levels of FXIII and α_2_AP were measured in the cryoprecipitate, RiaSTAP^®^ and FibCLOT^®^ using a commercially available ELISA from Abcam and Insight Biotechnology, following the manufacturer’s instructions. The concentrations of PAI-1 were determined using Simple Plex™ assays on the Ella™ system (ProteinSimple, Bio-techne, San Jose, CA, USA) following the manufacturer’s guidelines.

### 4.7. Quantra^®^ Haemostasis Analyser

Whole blood or 40% hemodiluted blood was spiked with cryoprecipitate, RiaSTAP^®^ or FibCLOT^®^ (1 mg/mL) and with 2.5 nM tPA and run on a QStat^®^ cartridge using a Quantra^®^ Hemostasis Analyzer (Hemosonics, Asnières-sur-Seine, France). The Quantra^®^ Hemostasis Analyzer measures the viscoelastic properties using SEER (Sonic Estimation of Elasticity via Resonance) Sonorheometry.

### 4.8. Chandler Model Thrombi

Chandler model thrombi were formed as previously described [44] with the following exceptions. Whole blood or 40% hemodiluted blood was spiked with cryoprecipitate, RiaSTAP^®^ or FibCLOT^®^ (1 mg/mL) and with Fluorescein isothiocyanate (FITC)-Fibrinogen (43.5 μg/mL). Clotting was initiated by CaCl_2_ (10.6 mM) under rotation for 90 min at 30 rpm at ambient temperature. After this time, thrombi were lysed in TBST containing 1 µg/mL tPA at 37 °C. Samples were taken every 30 min for 4 h, and fluorescence release was quantified in a BioTek FLx800 Fluorescence plate reader at excitation 485 nm and emission 528 nm using Gen5 software (Biotek Instruments, Winooski, VT, USA).

### 4.9. Statistical Analysis

The results are expressed as the mean and standard error of mean, where *n* ≥ 3. Statistical analysis was performed using GraphPad Prism Software (Versions 9.1, GraphPad Software Inc, La Jolla, CA, USA, Windows-based PC and Mac). Statistical significance was determined with a one-way ANOVA analysis of variance using Dunnett’s Post Hoc Comparison test, with a value of *p* < 0.05 being considered statistically significant. The clot turbidity data were analyzed using Shiny App Version 1.1 (https://drclongstaff.shinyapps.io/clotlysisCL_2019/, last accessed on 9 July 2021) [46].

## Figures and Tables

**Figure 1 ijms-23-02944-f001:**
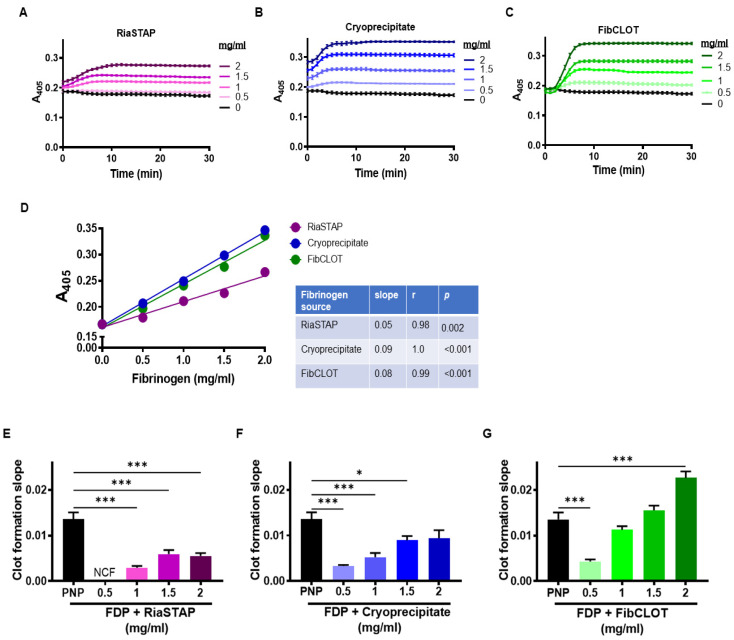
Supplementation of fibrinogen increases clot density. Clots contained 30% PNP or FDP, 16 µM phospholipids, ±FibCLOT^®^, RiaSTAP^®^ or cryoprecipitate at 0.5–2 mg/mL. Clotting was initiated with 0.1 U/mL thrombin and 10.6 mM CaCl2 and with a change in absorbance measured at 405 nm. (**A**–**C**) Absorbance curves. (**D**) Correlation of maximal absorbance with the concentration of fibrinogen. (**E**–**G**) Slope of clot formation, indicative of the rate of fibrin formation. Data shown are mean ± SEM, *n* = 4, * *p* < 0.05, and *** *p* < 0.001.

**Figure 2 ijms-23-02944-f002:**
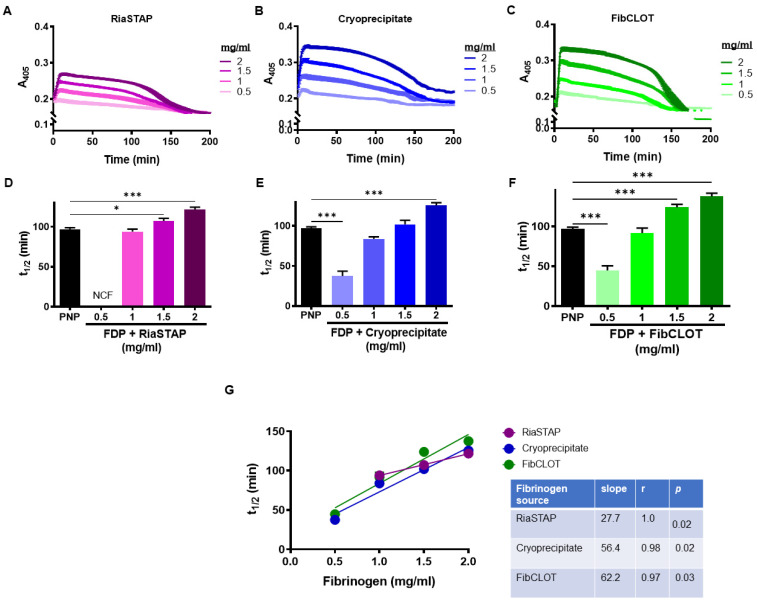
Addition of fibrinogen delays lysis. Clots contained 30% PNP or FDP, 16 µM phospholipids, 300 pM tissue plasminogen activator (tPA) ± FibCLOT^®^, RiaSTAP^®^ or cryoprecipitate at 0.5–2 mg/mL. Clotting was initiated with 0.1 U/mL thrombin and 10.6 mM CaCl_2_ and the change in absorbance measured at 405 nm. (**A**–**C**) Raw lysis curves and (**D**–**F**) 50% lysis time show a clear increase in resistance to lysis with increasing fibrinogen concentration of RiaSTAP^®^, cryoprecipitate and FibCLOT^®^, respectively. (**G**) Correlation of 50% clot lysis times with fibrinogen concentration. Data shown are mean ± SEM, *n* = 4, * *p* < 0.05 and *** *p* < 0.001 compared with PNP.

**Figure 3 ijms-23-02944-f003:**
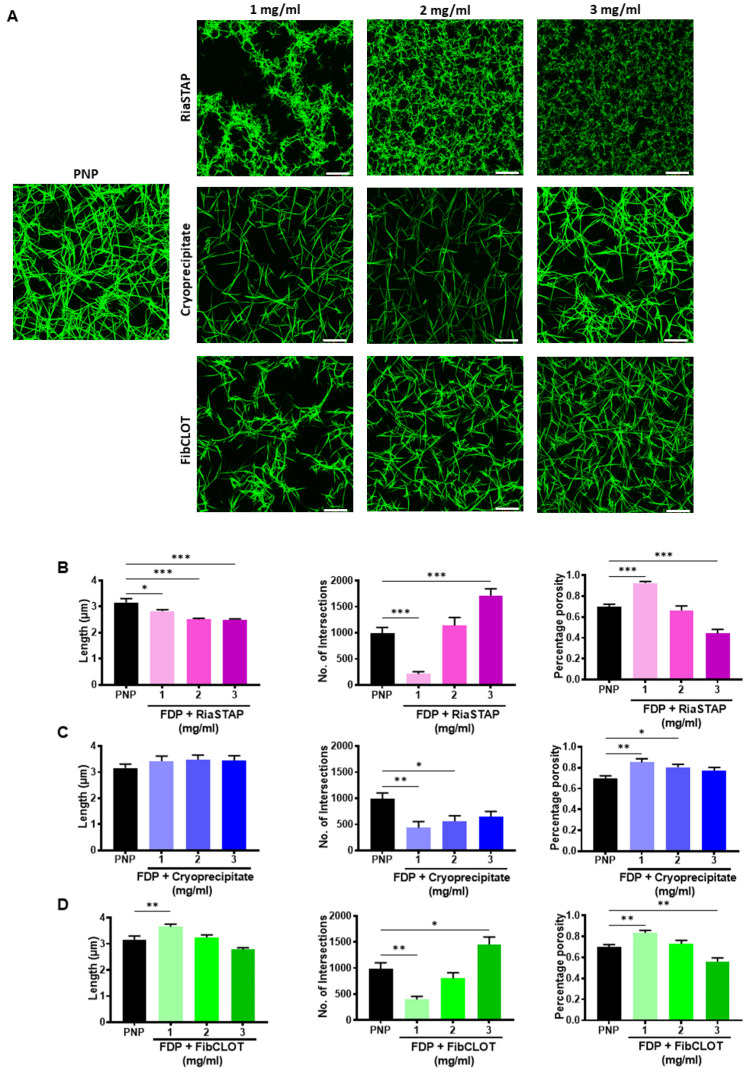
Impact of fibrinogen source on fibrin network structure. Clots were formed containing 30% PNP or FDP, 0.25 µM AF488 fibrinogen, 16 µM phospholipids ± RiaSTAP^®^, cryoprecipitate or FibCLOT. Clots were polymerized by addition of 0.1 U/mL thrombin and 10.6 mM CaCl_2_ for 2 h at 37 °C. Z-stack images were collected on an LSM 710 confocal microscope using a 63 X oil objective. Scale bar = 20 µM. (**A**) Images representative of *n* = 3. Data shown are the mean ± SEM fibrin fiber length (μm), number of intersections present indicating branching and the percentage porosity of the clots for (**B**) RiaSTAP^®^ (**C**) cryoprecipitate and (**D**) FibCLOT^®^. * *p* < 0.05, ***p* < 0.01 and *** *p* < 0.001.

**Figure 4 ijms-23-02944-f004:**
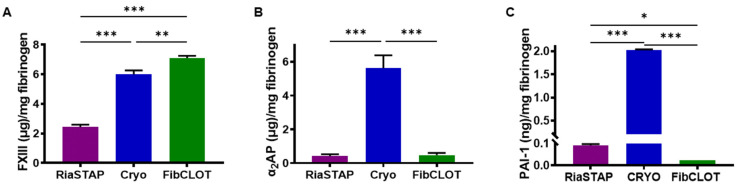
Quantification of FXIII, α_2_AP and PAI-1 levels in fibrinogen preparations. (**A**) FXIII, (**B**) α_2_AP and (**C**) PAI-1 in RiaSTAP^®^, cryoprecipitate or FibCLOT^®^ was measured by a commercially available ELISA. Data are expressed as mean protein concentration ± SEM detected per mg of fibrinogen. of *n* = 3, * *p* < 0.05, ** *p* < 0.01, *** *p* < 0.001.

**Figure 5 ijms-23-02944-f005:**
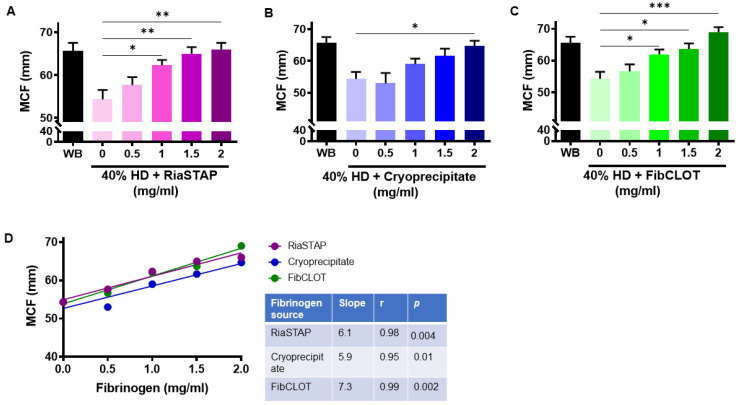
Clot strength is directly proportional to the concentration of fibrinogen supplementation. Thromboelastography was performed on whole blood (WB) or 40% hemodiluted clots (HD) ± (**A**) RiaSTAP^®^, (**B**) cryoprecipitate or (**C**) FibCLOT^®^ (0.5–2 mg/mL) using EXTEM reagent and analyzed using ROTEM thromboelastography. Data represent mean data ± SEM of maximum clot firmness (MCF). (**D**) Correlation of MCF with fibrinogen concentration. *n* = 3, * *p* < 0.05, ** *p* < 0.01, *** *p* < 0.001 vs. 0.5 mg/mL.

**Figure 6 ijms-23-02944-f006:**
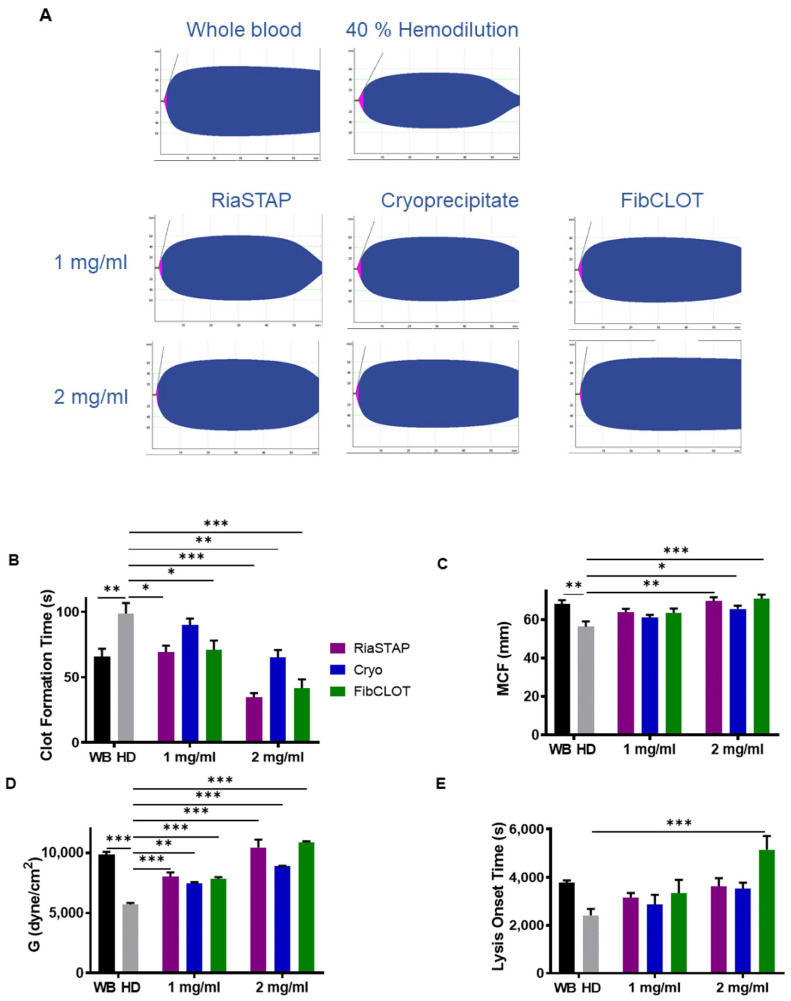
Effect of hemodilution on thromboelastography parameters in the presence of tPA. Thromboelastography was performed on whole blood or blood diluted by 40% (HD) + 0.4 nM tPA ± 1 or 2 mg/mL, RiaSTAP^®^ cryoprecipitate or FibCLOT^®^ using EXTEM reagent. (**A**) Representative thromboelastography traces, (**B**) clot formation time (CFT), (**C**) maximum clot firmness (MCF), (**D**) clot elasticity and (**E**) lysis onset time. * *p* < 0.05, ** *p* < 0.01, *** *p* < 0.001, vs. HD. *n* = 3.

**Figure 7 ijms-23-02944-f007:**
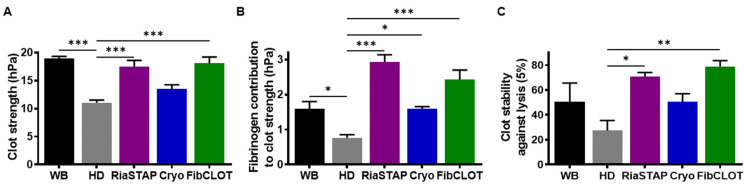
Clots formed after hemodilution are stabilized by fibrinogen supplementation. Whole blood or 40% hemodiluted (HD) blood spiked with cryoprecipitate, RiaSTAP^®^ or FibCLOT^®^ (1 mg/mL), and tPA (2.5 nM) was run on a QStat^®^ cartridge using a Quantra^®^ Haemostasis Analyzer. (**A**) Clot strength, (**B**) fibrinogen contribution to clot strength and (**C**) clot stability against lysis. Data represent mean ± SEM * *p* < 0.05, ** *p* < 0.01, *** *p* < 0.001, vs. HD. *n* = 3.

**Figure 8 ijms-23-02944-f008:**
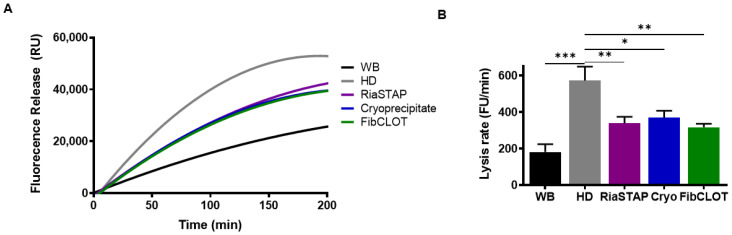
Fibrinogen supplementation stabilizes thrombi formed at hemodilution against fibrino lysis. (**A**) FITC-fibrinogen (43.5 µg/mL) was added to whole blood (black) or 40% HD blood in the presence of 1 mg/mL FibCLOT^®^, RiaSTAP^®^ or cryoprecipitate. The thrombi were formed by recalcification (10.9 mM CaCl_2_) and rotated on at 30 rpm for 90 min. The thrombi were lysed with tPA (1 µg/mL), and lysis was measured as fluorescence release of degradation products. (**B**). The thrombus lysis rate was calculated by change in fluorescence unit (FU) per minute. * *p* < 0.05, ** *p* < 0.01, *** *p* < 0.001 vs. HD. *n* = 3.

## Data Availability

The data presented in this study are available on request from the corresponding author.

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
