# Peer review of "The Efficacy of Fibrinogen Concentrates in Relation to Cryoprecipitate in Restoring Clot Integrity and Stability against Lysis"

_ijms, 2022, doi:10.3390/ijms23062944_

Round 1
Reviewer 1 Report
This paper compared the efficiency of cryoprecipitate and the fibrinogen concentrates, RiaSTAP and FibCLOT, in restoring clot integrity in models of TIC. Cryoprecipitate and FibCLOT produced clots with higher maximal absorbance and enhanced resistance to lysis relative to RiaSTAP. This work highlights differences in the efficacy of these two fibrinogen concentrates analyzed, presumably arising due to differences in their preparation and composition.
It shows that 1 mg/ml fibrinogen, a clinically achievable concentration, can restore adequate clot integrity and Fib-CLOT was superior in normalizing clot structure and stabilizing haemodilution clots against mechanical and fibrinolytic degradation. This topic would be beneficial to the community. However, more careful proofreading of the current manuscript is recommended as multiple grammar errors/spelling mistakes are found in the draft.
Author Response
"Please see the attachment."

Reviewer 2 Report
The authors describe a very interesting original data, which is focused on traumatic coagulopathy therapy and comparison of fibrinogen concentrates and cryoprecipitate.
The scope of the manuscript is reasonable, the authors do well in staying focused and on key point determine differences in clot strength and stability between cryoprecipitate and fibrinogen concentrate. The authors do a solid job. Text is supported by several charts / tables and figures, which substantively add to the manuscript. The comparison of individual fibrinogen concentrates is interesting, but it should be noted that these are plasma derivatives that have very similar properties and the differences between the individual concentrates are minimal. Among other things, most fibrinogen concentrates are used in off label indications.
The manuscript is well structured, but some parts are missing some important facts that authors should add.
Page 1, lines 37-38: In this section, the authors should also provide a reference in the case describing the case of perioperative management in a patient with a fibrinogen disorder, where fibrinogen values of 1.5 g / l are important for maintaining hemostasis without bleeding on the first days postoperatively. Semin Thromb Hemost. 2016 Sep;42(6):689-92. doi: 10.1055/s-0036-1585079.
Page 2, lines 57-58 Fibrinogen concentrates are used primarily several Western European countries, but cryoprecipital is still administered in the United States and the United Kingdom. In the part, the authors should clearly state the important fact that a standard dose of 10-20 units (500-1000 mL) of cryoprecipitate is awaited to increase fibrinogen activity by 0.6-1.2 g / L in a 70 kg adult. These facts were published in a manuscript that the authors should cite. Diagnostics 2021, 11(11), 2140; https://doi.org/10.3390/diagnostics11112140
Discussion: the beginning of the discussion, the authors should state that fibrinogen replacement is used not only in bleeding in acute medicine, but also in the prevention of congenital coagulopathies. . This sentence should also include a reference in which fibrinogen replacement is crucial in congenital coagulopathy - afibrinogenemia in a prophylactic regimen in patients with recurrent bleeding. It is appropriate to cite the following manuscript, which described the benefits of prophylaxis in congenital afibrinogenemia Blood Coagul Fibrinolysis. 2015 Dec;26(8):978-80. doi: 10.1097/MBC.0000000000000392.
The methodological part is processed very thoroughly.
Figures in the text are very clearly written.
I have to say that with these 42 references. Only 10 references are in the last 5 years. Authors should also add newer references.
Author Response
"Please see the attachment."

Round 2
Reviewer 2 Report
The presented manuscript has been corrected in response to the suggestions. The authors have followed the recommendations of the reviewer. After the revision, the provided data and addition of the results became more clear. I would like to thank the authors for resubmitting the manuscript and explaining the obscure points from the previous version.